# LEARNING A DISENTANGLING REPRESENTATION FOR PU LEARNING

## ABSTRACT

In this paper, we address the problem of learning a binary (positive vs. negative) classifier given Positive and Unlabeled data commonly referred to as PU learning. Although rudimentary techniques like clustering, out-of-distribution detection, or positive density estimation can be used to solve the problem in low-dimensional settings, their efficacy progressively deteriorates with higher dimensions due to the increasing complexities in the data distribution. In this paper we propose to learn a neural network-based data representation using a loss function that can be used to project the unlabeled data into two (positive and negative) clusters that can be easily identified using simple clustering techniques, effectively emulating the phenomenon observed in low-dimensional settings. We adopt a vector quantization technique for the learned representations to amplify the separation between the learned unlabeled data clusters. We conduct experiments on simulated PU data that demonstrate the improved performance of our proposed method compared to the current state-of-the-art approaches. We also provide some theoretical justification for our two cluster-based approach and our algorithmic choices.

## 1 INTRODUCTION

The excessive data demands of current large deep learning models can make the cost of data collection and labeling prohibitive. These costs along with other challenges related to data collection have given rise to the development of learning settings that deal with data scarcity and the absence or poor quality of labeling. PU learning, or learning from Positive and Unlabeled data is a learning setting that deals with binary classification problems where the labeling of one of the two classes is either significantly costly or even infeasible (Bekker & Davis, 2020). This scenario exists naturally in many problems such as medical diagnosis where a single clear symptom of a disease can be reliably used as an indicator of the "diseased" patients (positive class). However, the absence of this symptom does not conclusively rule out the existence of the disease. Consequently, the data will contain a positive label for those cases that exhibit that symptom, while all other cases will remain unlabeled (Claesen et al., 2015). Another clear example of a PU setting is seen in spam detection, where it is usually easy to label emails reported by users as spam (positive class), while the label of all other emails remains unknown (Wu et al., 2018). PU learning appears in other fields including matrix completion (Hsieh et al., 2015), gene identification (Mordelet & Vert, 2011), and recommendation systems (Zhou et al., 2021).

Many existing PU learning methods (such as Weighted Unlabeled Samples SVM (Liu et al., 2008), Biased Least Squares SVM (Ke et al., 2018), Topic-Sensitive pLSA (Zhou et al., 2009), and Rank Pruning (Northcutt et al., 2017)) primarily leverage the bi-modality of the unlabeled data distribution. The bi-modality arises from the distributional contrast between positive and negative samples within the unlabeled dataset. These methods, although effective in some contexts where the bi-modality of the unlabeled data is easily identifiable, show a gradual decline in performance as the dimensionality of the data increases and the positive and negative instances within the unlabeled data become entangled and less distinguishable.

The existence of the PU learning problem in domains where the dimensionality of the data is high has led to the emergence of learning methods that deal with the problem in more subtle ways, such as: (i) using generative models to learn the negative data distribution to reduce the problem to a supervised learning scenario (Chiaroni et al., 2020) (Zamzam et al., 2023), (ii) two step methods that estimate

the positive class prior and then utilize it to learn a binary classifier (Garg et al., 2021) (Elkan & Noto, 2008), and (iii) adversarial learning where two classifiers iteratively learn the separation between positive and negative instances (Hu et al., 2021). Nevertheless, the intricate entanglement of the two classes within the unlabeled data in high dimensions continues to exert a significant influence on the performance trajectories of these methods across diverse scenarios.

One simple and yet unexplored way of dealing with the complexity of the data in high dimensions is to learn a new representation that makes the distributional difference between positive and negative instances within the unlabeled data easily identifiable. We propose a unique representation learning method that projects the positive and unlabeled data into a new space where the unlabeled data gets disentangled into two separable clusters; one of these clusters coincides with the representation of the positive labeled samples, and the other is recognized as the representation of the negative samples, replicating the separability phenomenon found in lower-dimensional spaces. **The contributions of this paper are outlined as follows:**

- A novel loss function designed to facilitate the learning of a new data representation in which the unlabeled data disentangles into two distinct positive and negative clusters.

- An innovative adoption of vector quantization techniques to enhance the informative capacity of the learned representation, particularly in the context of PU learning, bridging the performance gap between existing PU learning methods and traditional supervised learning.

- Empirical evidence of the effectiveness of our proposed method through experimental studies using four different datasets.

- Comprehensive ablation studies that emphasize the significance and impact of each term within our loss function. These studies also showcase the method's robustness across a wide range of hyperparameter configurations.

- We also provide some theoretical justification for our two cluster-based approach and some of our algorithmic choices.

## 2 PROBLEM SETUP

In PU learning, the goal is to learn a binary (positive vs. negative) classifier given labeled positive data and unlabeled data that consists of positive and negative samples. We propose to achieve this by first learning a new data representation in which the positive and negative samples become more distinguishable, and then deploying a simple clustering technique to learn the two classes.

To formalize the PU problem setup, we denote the class-conditional distributions for the positive and negative classes by $\mathcal{P}_P$ and $\mathcal{P}_N$, where $p_P(x) = p(x|y = 1)$ and $p_N(x) = p(x|y = 0)$ represent their respective class-conditional densities, and the distribution of the unlabeled data by $\mathcal{P}_U$, with $p_U(x) = p(x)$ denoting its density. We also denote by $\alpha$ the proportion of positive samples within the unlabeled distribution ($\alpha = p(y = 1)$).

In this setting, a set of $n_p$ independent and identically distributed (i.i.d.) samples is drawn from the positive class conditional distribution, resulting in $\mathcal{X}_P = \{x_1, x_2, ..., x_{n_p}\} \sim \mathcal{P}_P^{n_p}$, where each $x_i \in \mathbb{R}^d$. Similarly, the unlabeled set $\mathcal{X}_U$ is partitioned into two subsets: $\mathcal{X}_{UP}$ containing $n_{up}$ positive samples and $\mathcal{X}_{UN}$ containing $n_{un}$ negative samples, resulting in $\mathcal{X}_U = \mathcal{X}_{UP} \cup \mathcal{X}_{UN}$, where each $x_i \in \mathbb{R}^d$. We do not assume a known positive class prior $\alpha$, and our goal is to learn a new representation space in which the Euclidean distances between the samples within $\mathcal{X}_{UN}$ and within the union $\mathcal{X}_{UP} \cup \mathcal{X}_P$ are minimized, while simultaneously maximizing the distances between samples across $\mathcal{X}_{UN}$ and $\mathcal{X}_{UP} \cup \mathcal{X}_P$, resulting in an easily identifiable separation between positive and negative samples.

## 3 LEARNING A DISENTANGLING REPRESENTATION FOR PU LEARNING

We start by introducing a motivation to solve the problem of PU learning through learning a new representation space. In the toy 1-dimensional example shown in Figure 1, the PU problem setting is simulated using two Gaussian distributions with means 0 and 30 and variances of 9 and 25 for the positive and negative classes, respectively. The unlabeled set is a combination of samples that

come from both classes with equal probabilities. The clear difference between the two modes in the bimodal distribution of the unlabeled data allows a simple K-means algorithm deployed only on the unlabeled data to learn the two underlying positive and negative classes. The positive samples can then be used to identify which of the two learned clusters corresponds to the positive class by measuring the distance between the positive samples and the centers of the two learned clusters. This simple example illustrates how in low-dimensional settings (mainly because of the obvious difference between the modes in the unlabeled data) the problem is easily solvable.

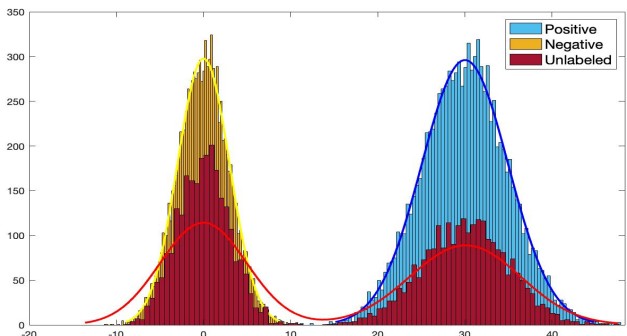

Figure 1: 1D toy example showing the distributions of positive, negative, and unlabeled sets in a PU learning problem.

Driven by the evident simplicity of the problem when the two modes in the unlabeled data are readily distinguishable, we present an approach to learn a representation that tackles scenarios where these two modes are challenging to differentiate. Although multiple existing PU learning methods have been proposed to deal with the complexities of high-dimensional data, we show that learning a new representation alleviates the problem and gives consistent results across different domains. Figure 2 shows a simple comparison between the t-SNE visualization of the unlabeled data in the representation space in our proposed method and in the representation space of a classical VQ-VAE (Van Den Oord et al., 2017). The figure clearly shows the effectiveness of the proposed method in learning a representation space in which the positive and negative samples are clearly concentrated in two clusters, making the problem much closer to the simple 1-dimensional scenario shown in Figure 1. The representation of the unlabeled data in the VQ-VAE shows the entanglement of the positive and negative samples, which makes it challenging to learn a binary classifier in the PU setting.

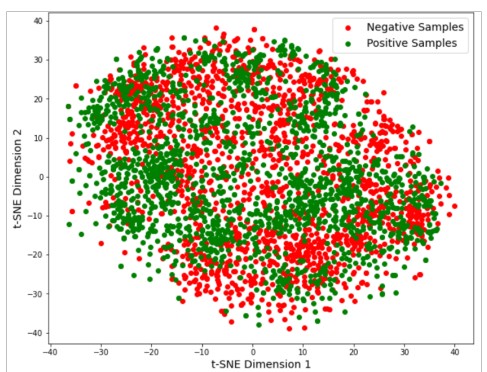
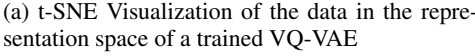
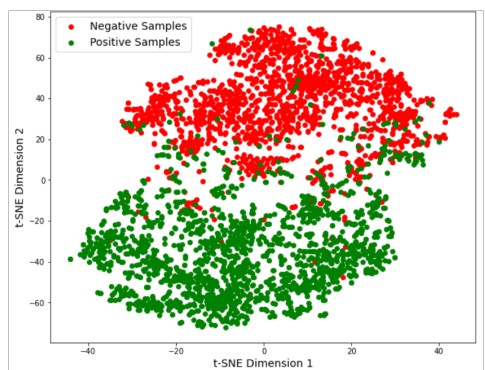

(a) t-SNE Visualization of the data in the representation space of a trained VQ-VAE

(b) t-SNE Visualization of the data in the representation space of our proposed VQ-Encoder

Figure 2: The t-SNE visualization of the learned data representation (for the AFHQ dataset) in the proposed method compared to the data representation learned in a classical VQ-VAE showing how the proposed learned representation disentangles the positive and negative samples such that they can be easily told apart using simple clustering algorithms.

### 3.1 METHODOLOGY

Let $f_\theta : \mathbb{R}^d \mapsto \mathbb{R}^{K \times p}$ represent a network that transforms input data from the input space into a set $\mathbf{V}$ of $K$ vectors $\mathbf{V} = \{\mathbf{v}_1, \mathbf{v}_2, \ldots, \mathbf{v}_K\}$, where $\mathbf{v}_i \in \mathbb{R}^p$. Consider a codebook $\mathbf{C} = \{\mathbf{c}_1, \mathbf{c}_2, \ldots, \mathbf{c}_m\}$ of $m$ vectors, where $\mathbf{c}_i \in \mathbb{R}^p$, such that $\|\mathbf{c}_1\|_2 < \|\mathbf{c}_2\|_2 < \ldots < \|\mathbf{c}_m\|_2$. We define a quantization operator $Q(\cdot)$ whose output is defined as $Q(\mathbf{v}) = \underset{\mathbf{c}_k}{\arg\min} \|\mathbf{v} - \mathbf{c}_k\|_2$.

We propose to minimize the following loss function:

$$\mathcal{L}(\theta) = \sum_{i_p=1}^{n_P} \sum_{j=1}^{K} \|\mathbf{v_j}(\mathbf{x_{i_p}}; \theta) - sg(\mathbf{c}_m)\|_2^2 + \|sg(\mathbf{v_j}(\mathbf{x_{i_p}}; \theta)) - Q(\mathbf{v_j}(\mathbf{x_{i_p}}; \theta))\|_2^2$$

$$+ \sum_{i_u=1}^{n_u} \sum_{j=1}^{K} \|\mathbf{v_j}(\mathbf{x_{i_u}}; \theta) - sg(\mathbf{c}_1)\|_2^2 + \|sg(\mathbf{v_j}(\mathbf{x_{i_u}}; \theta)) - Q(\mathbf{v_j}(\mathbf{x_{i_u}}; \theta))\|_2^2 \quad (1)$$

Where $sg(\cdot)$ is the stop gradient operator that stops the gradient from being propagated back to its operand during backpropagation, making it a constant non-updated value, $\mathbf{v_j}(\mathbf{x_{i_p}})$ is the $j$'th vector in the output of the encoder network after inputting the $i_p$'th sample from the positive set $X_P$, and $\mathbf{v_j}(\mathbf{x_{i_u}})$ is the $j$'th vector in the output of the encoder network after inputting the $i_u$'th sample from the unlabeled set $X_U$.

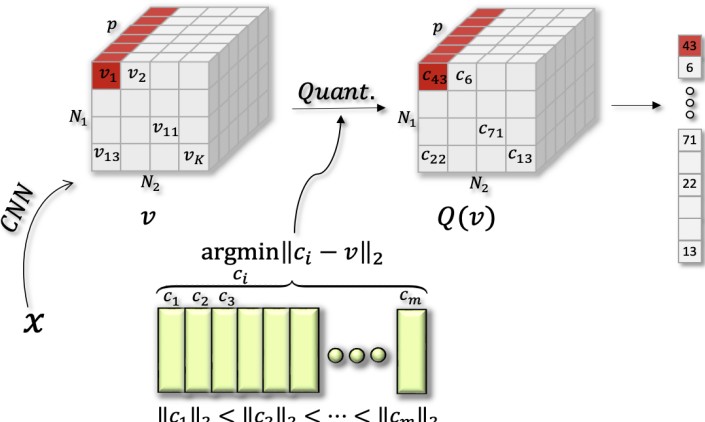

Figure 3: Illustration of the proposed vector quantized encoder: The input image $x$ is fed to a Convolutional Neural Network that encodes it to $K = N_1 \times N_2$ vectors of dimension $p$. The quantization operator $Q(\cdot)$ is applied to the $n$ vectors, resulting in $n$ codebook vectors whose indices are outputted as the new representation to be used by a K-mean clustering algorithm.

The first term in the two summations is to derive the vectors in the encoding of the positive samples towards the codebook vector of the highest magnitude $\mathbf{c}_m$, and derive the vectors in the encoding of the unlabeled samples towards the codebook vector of the lowest magnitude $\mathbf{c}_1$. The second term in the two summations is to update the codebook vectors to align with the output of the encoder for faster convergence, inspired by (Van Den Oord et al., 2017).

The main idea in the loss function is that encoding the positive samples in the (labeled) positive set to a vector of high magnitude and the positive samples in the unlabeled set to a vector with a low magnitude would intuitively result in projecting all the positive samples to a linear combination of the two vectors that depends on the proportion of the positive samples in the unlabeled set. On the other hand, the negative samples in the unlabeled set will simply be projected to the vector of the low magnitude as it's the only vector they're projected to in the loss function. The similarity in distribution between the positive labeled samples and the positive unlabeled samples is the main feature that is being exploited here, deliberately making the network unable to project each of them to its corresponding vector in the loss function while being able to more easily project the negative

samples to a different vector, creating the sought-after separation between the positive and negative samples in the encoding space.

After arriving at the new desired quantized representation, a K-means algorithm is employed on the indices of the vectors $Q(\mathbf{v})'s$ for all the samples in the unlabeled set to cluster them into 2 clusters. The K-means algorithm returns an assigned cluster for each of the unlabeled instances and the center of the two learned clusters. The positive labeled samples are then encoded and the distance from their encodings are compared to the two centers returned by the K-means algorithm, and the closest cluster to the positive labeled samples encodings are recognized to be the positive cluster. At inference time, the two centroids returned by the K-means algorithm are compared to the encoding of a test sample to decide its corresponding class based on its Euclidean proximity to the two centers.

## 3.2 MATHEMATICAL INTUITION

While encoding the input to a set of vectors and using vector quantization are both shown to improve the performance of the proposed method (see in detail analysis in the ablation studies in section 5.5), the mathematical basis of the proposed method can be simply explained in a scenario where the encoder network encodes the input data $x$ to a single vector $v(x;\theta)$. In what follows, we show the mathematical logic behind the proposed method in this single vector scenario. Consider an encoder network that maps the input from $\mathbb{R}^d$ to $\mathbb{R}^p$. Consider $\mu_P, \mu_U \in \mathbb{R}^p$ where $\mu_P \neq \mu_U$. Suppose the following loss function is to be optimized

$$\min_\theta \bar{\mathcal{L}}(\theta) = \min_\theta \; \mathbb{E}_{x^p \sim \mathcal{P}_P}[\|v(x^p;\theta) - \mu_P\|_2^2] + \mathbb{E}_{x^u \sim \mathcal{P}_U}[\|v(x^u;\theta) - \mu_U\|_2^2] \tag{2}$$

$$= \min_\theta \; \mathbb{E}_{x^p \sim \mathcal{P}_P}[\|v(x^p;\theta) - \mu_P\|_2^2] + \alpha\mathbb{E}_{x^{up} \sim \mathcal{P}_P}[\|v(x^{up};\theta) - \mu_U\|_2^2]$$
$$+ (1-\alpha)\mathbb{E}_{x^{un} \sim \mathcal{P}_N}[\|v(x^{un};\theta) - \mu_U\|_2^2] \tag{3}$$

$$= \min_\theta \; \mathbb{E}_{x^p \sim \mathcal{P}_P}\left[\|v(x^p;\theta) - \mu_P\|_2^2 + \alpha\|v(x^p;\theta) - \mu_U\|_2^2\right]$$
$$+ (1-\alpha)\mathbb{E}_{x^{un} \sim \mathcal{P}_N}\left[\|v(x^{un};\theta) - \mu_U\|_2^2\right] \tag{4}$$

Replacing the expectations in (4) with the empirical average results in:

$$\min_\theta \bar{\mathcal{L}}(\theta) = \min_\theta \frac{1}{n_p}\sum_{i=1}^{n_p}[\|v(x_i^p;\theta) - \mu_P\|_2^2] + \frac{\alpha}{n_{up}}\sum_{i=1}^{n_{up}}[\|v(x_i^{up};\theta) - \mu_U\|_2^2]$$
$$+ \frac{1-\alpha}{n_{un}}\sum_{i=1}^{n_{un}}[\|v(x_i^{un};\theta) - \mu_U\|_2^2] \tag{5}$$

Under the assumption that the labeled positive samples are Selected Completely At Random (SCAR (Elkan & Noto, 2008)), i.e., there is no difference between the distribution of the labeled and unlabeled positive samples, the loss function in equation 5 is minimized at the linear combination $v(x^p;\theta) = v(x^{up};\theta) = \frac{\mu_P + \alpha\mu_U}{1+\alpha}$ and $v(x^{un};\theta) = \mu_U$. For $\mu_P$ and $\mu_U$ sufficiently distant from each other (in the $\|.\|_2$ sense), a simple K-means algorithm can be used to cluster $v(x;\theta)$ for positive and negative $x$'s.

To make this intuition concrete we state an informal theorem below which we make more precise in the appendix.

**Theorem 1** (Informal). *Consider the formulation in equation 5 and assume that all the layers of the neural network $x \mapsto g(x;\theta)$ are sufficiently wide (large number of channels). We run gradient updates on the loss equation 5 with an appropriate choice of step size $\eta$ starting from random initialization. We assume that the scale of initialization of the network is sufficiently large (i.e. the standard deviation of the weights at initialization). Then, with early stopping at a time $T$ (specified in the appendix) we have*

$$v(x_i^p;\theta_T) = v(x_i^{up};\theta_T) \approx \frac{\mu_P + \alpha\mu_U}{1+\alpha} \quad and \quad v(x_i^{un};\theta_T) \approx \mu_U$$

*holds for all $i$ with high probability.*

We note that the above theorem holds for a rather broad range of step sizes (see Appendix A for a precise description).

Although the loss function in equation 5 is written in terms of $\alpha$, $n_{up}$ and $n_{un}$ which are assumed unknown in this work, it is easy to see that using the empirical estimates of $\alpha \approx \frac{n_{up}}{n_u}$ and $1 - \alpha \approx \frac{n_{un}}{n_u}$, we have $\frac{\alpha}{n_{up}} = \frac{1-\alpha}{n_{un}} \approx \frac{1}{n_u}$ which results in

$$
\min_\theta \bar{\mathcal{L}}(\theta) = \min_\theta \frac{1}{n_p} \sum_{i=1}^{n_p} [\|v(x_i^p; \theta) - \mu_P\|_2^2] + \frac{1}{n_u} \sum_{i=1}^{n_{up}} [\|v(x_i^{up}; \theta) - \mu_U\|_2^2]
$$
$$
+ \frac{1}{n_u} \sum_{i=1}^{n_{un}} [\|v(x_i^{un}; \theta) - \mu_U\|_2^2] \qquad (6)
$$
$$
= \min_\theta \frac{1}{n_p} \sum_{i=1}^{n_p} [\|v(x_i^p; \theta) - \mu_P\|_2^2] + \frac{1}{n_u} \sum_{i=1}^{n_u} [\|v(x_i^u; \theta) - \mu_U\|_2^2]
$$

This allows the optimization of the loss in equation 5 without the need of any knowledge about $\alpha$.

### 3.3 STOPPING CRITERIA

One major difference between supervised learning and learning from PU data is the absence of any completely labeled validation sets. Consequently, there is no obvious metric that can be used in general to avoid overfitting. Some existing PU learning methods design a model such that it eventually converges to the correct answer after exhaustive training (Garg et al., 2021), (Zamzam et al., 2023). One significant disadvantage of these methods is that the speed of convergence is unknown, hence, there is no definitive way of determining a reasonable stopping point. As a result, to increase confidence in the correctness of the solution, one has to train the model for a large number of epochs.

In this study, the K-means algorithm is the primary model utilized to differentiate between positive and negative samples. Consequently, the outputs of the K-means algorithm (applied on the unlabeled data during training) are used to identify the overfitting behavior. By the design of the loss function, the unlabeled samples are driven toward one vector, and the positive samples are driven toward another vector, relying on the challenge introduced to the encoder network in differentiating between labeled and unlabeled positive samples, hence, they are projected to a linear combination of the vectors. An overfitting encoder network would start memorizing the labeled and unlabeled positive samples, projecting each of them to the corresponding vector in the loss function. Since the K-means algorithm is applied on the unlabeled data, the centers of the clusters identified by the algorithm will start getting closer to each other as the unlabeled data gets memorized and dealt with by the network in the same way. These centers identified by the K-means algorithm are monitored during the training and the training of the encoder network is stopped once the distance between the two centers starts decreasing.

## 4 RELATED WORK

The PU learning problem has been discussed in the literature for at least 25 years. More recently due to the growing data requirements of machine learning and deep learning models, various approaches have been developed to address the issue in different fields where the labeling of one or more classes is impractical or costly ((Liu et al., 2003), (Yu et al., 2004), (Zhang & Lee, 2005), (Elkan & Noto, 2008), (Zamzam et al., 2023) (Hsieh et al., 2015), (Chiaroni et al., 2020),(Garg et al., 2021), (Zhao et al., 2022)).

A common approach to deal with the PU learning problem is to consider the unlabeled observations as belonging to the negative class and dealing with their labels as noisy labels. To accomplish this, a binary classifier is trained using a biased cost function that places a higher penalty for the misclassification of positive samples compared to that of the unlabeled (noisy negative) samples (Liu et al., 2003), (Hsieh et al., 2015), (Mordelet & Vert, 2014). Another related class of methods assumes a known prior probability for the positive class $P(Y = 1)$. By incorporating this known class prior, the bias in the cost function can be accurately weighted towards the positive class. Alternatively, one can train a binary classifier by assuming that only a subset of unlabeled samples with the lowest

loss values are reliable negative samples. The number of samples chosen in the subset must ensure that the proportion of the remaining samples in the unlabeled set is equal to the positive class prior (Kiryo et al., 2017) (Zhao et al., 2022) (Plessis et al., 2015). The main disadvantage of this class of methods is that in practice, the positive class prior is rarely known. To overcome this, a family of methods has been proposed to solve the problem by estimating the positive class prior as a first step, and subsequently, a classifier is trained using this information (Ivanov, 2020). Alternating between the step of estimating the prior and training the binary classifier has also been used in (Garg et al., 2021) ($TED^n$). Another class of PU learning methods defines a distance metric to identify the unlabeled observations that are the furthest from the positive samples. These observations are then treated as reliable negatives, reducing the problem to the supervised setting where both reliable negative and positive samples are available (Yu et al., 2004) (Grinenko et al., 2018). An alternative way of finding reliable negative examples is to generate them using a Generative Adversarial Network (GAN) (Chiaroni et al., 2020) (Zamzam et al., 2023), similarly, reducing the problem to a supervised setting. However, generating negative samples and relying on them to train a classifier in an supervised way often shows deterioration of the performance as the complexity of the data increases.

Here we propose a method to use the unlabeled and positive data to train an encoder that learns to encode the data to a representation space, where the positive and negative samples are distant enough from each other to be identified using a K-means algorithm. After applying the K-means algorithm to the unlabeled data, we used the labeled positive samples to determine which of the two resulting clusters corresponds to the positive class. Details of the proposed method and empirical comparisons with other methods are presented below.

## 5 EXPERIMENTS

### 5.1 USED DATASETS

We use 4 different datasets to evaluate the performance of the proposed method, namely MNIST (Deng, 2012), Fashion-MNIST (Xiao et al., 2017), CIFAR-10 (Krizhevsky, 2009), and animal faces (AFHQ) (Choi et al., 2020). The positive and negative classes are defined respectively as the last five classes vs. first five classes on Fashion-MNIST (classes: T-shirt, Trouser, Pullover, Dress, Coat, Sandal, Shirt, Sneaker, Bag, and Ankle boot), animal versus not animal images on CIFAR-10, even versus odd digits on MNIST dataset, and cat versus dog images on AFHQ.
We construct the training dataset $X = \{x_1, ..., x_p, x_{p+1}, ..., x_{p+n}\}$, consisting of $p$ positive samples, and $n$ negative samples. We randomly sample $\alpha|X_U|$ samples from the positive samples along with $(1 - \alpha)|X_U|$ negative samples to constitute the unlabeled set, where $\alpha$ is the proportion of positive samples in $X_U$, and $|X_U|$ is the size of $X_U$. We use the same data splits as in (Zamzam et al., 2023) to compare the different methods.

### 5.2 BASELINE METHODS

We compare our method to three state-of-the-art PU learning methods that have shown good performance on image datasets. The first method is *Observer-GAN* (Zamzam et al., 2023), which uses a GAN-based setup to train a classifier to learn features from the positive and unlabeled data that can be used to differentiate between positive and negative samples. The second is $TED^n$(Garg et al., 2021), which uses an alternating procedure between the problem of estimating the positive prior $\alpha$, and the problem of learning a binary classifier. The third method, *D-GAN* (Chiaroni et al., 2020), uses a two-step approach: in the first step, the generator network in a GAN is trained to generate pseudo-negative samples, and in the second step, a binary classifier is trained on the positive samples and the generated pseudo-negative samples. Since the first two methods claim convergence to the correct solution, and there is no clear way to stop training at an early stage, we follow the same method of training each of the methods for 1000 epochs. We then look at the average performance of the last 50 and 100 epochs. For the third method, no specific criteria were presented for terminating the training of the second-stage classifier. Therefore, we train the classifier and apply early stopping based on a fully labeled validation set to prevent the second-stage classifier from overfitting.
For our proposed method, we look at the Euclidean distance between the two clusters identified by the K-means algorithm (applied to the training set), and stop training when this distance starts decreasing. Figure 4 shows that even though the accuracy on the validation set does not change

dramatically after it reaches about 20 epochs, the point at which the distance between the two clusters found using the training set is largest also corresponds to the point of highest accuracy for the validation data. This behavior was evident in all experiments.

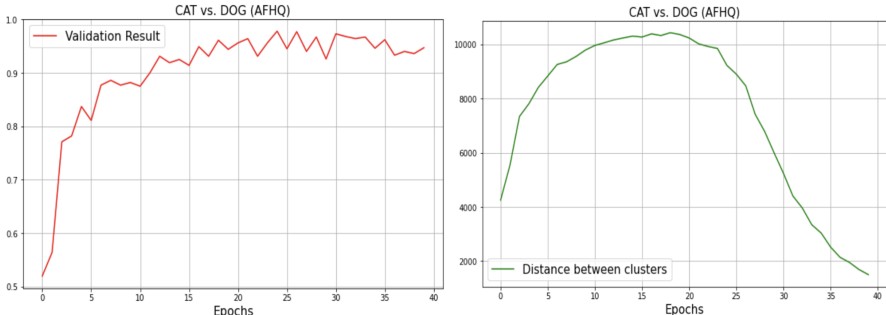

Figure 4: Left: Accuracy curve as a function of the number of epochs on the test data. Right: The Euclidean distance between the centers of the two clusters identified by the K-means algorithm on the unlabeled (training) data.

## 5.3 TRAINING SETUP

We utilized a neural network with six convolutional layers for all datasets. We choose the dimension of each output $v_i(x)$ and codebook vector to be 64. The number of the codebook vectors was 512, and they were all initialized from a normal distribution $\mathcal{N}(\underline{0}, I)$, where $\underline{0}$ is the vector of all zeros. We trained the proposed method on the loss function $\mathcal{L}$ specified in equation 1. We use Adam as the optimization algorithm with a learning rate of $10^{-4}$. We adopted the training specifications outlined in the original publications for all baseline methods.

## 5.4 COMPARISONS OF THE RESULTS

Table 2 shows the testing accuracy of each of the considered models on each of the datasets. The proposed method (VQ K-means) shows superior performance compared to all baseline methods.

In Table 1 we compare the number of epochs needed by each of the methods to reach 90% of its maximum attained accuracy when trained on the (AFHQ) dataset. The table shows that the simplicity of the proposed method (VQ K-means) allows for faster learning compared to other methods. We excluded the number of epochs for *D-GAN* from the table since it did not outperform random chance on this particular dataset.

Table 1: Number of epochs needed to reach 90% of the highest accuracy

| **Method** | $TED^n$ | *Observer* | **VQ K-means** |
|---|---|---|---|
| number of epochs | 43 | 220 | 11 |

| | D-GAN | $TED^n$ | | *Observer* | | VQ K-means |
|---|---|---|---|---|---|---|
| | **Early Stop** | **50** | **100** | **50** | **100** | **highest distance** |
| **AFHQ (Cats vs. Dogs)** | $50.3 \pm 0.2$ | $86.8 \pm 12$ | $89.9 \pm 14.9$ | $91 \pm 1.1$ | $90.1 \pm 3.2$ | $95.3 \pm 1.3$ |
| **CIFAR (Animal vs. Not Animal)** | $82 \pm 1.1$ | $88 \pm 2.5$ | $87.7 \pm 4.6$ | $89.6 \pm 0.7$ | $88.8 \pm 1.7$ | $91.1 \pm 0.7$ |
| **MNIST (Even vs. Odd)** | $98.3 \pm 0.1$ | $97.7 \pm 0.4$ | $97.7 \pm 0.4$ | $98.3 \pm 0.2$ | $97.8 \pm 1.6$ | $98.1 \pm 0.1$ |
| **Binarized Fashion MNIST** | $89.6 \pm 0.2$ | $88.5 \pm 0.9$ | $88.1 \pm 1$ | $92.6 \pm 0.3$ | $92 \pm 1$ | $93.3 \pm 0.75$ |

Table 2: Summary of experimental results averaged over 5 trials: Left-most column is the dataset, and upper-most row is the method used. Stopping Criteria: We report the best performing model when using *D-GAN*, the mean and standard deviation of the accuracy (%) of the last 50 and 100 epochs when using $TED^n$ or the *Observer* network, and the mean and standard deviation of the accuracy of the 5 models corresponding to the largest Euclidean distances between the centers of the clusters identified by K-means clustering of the unlabeled training data.

## 5.5 ABLATION STUDY

We empirically study various adaptations of the proposed method to evaluate the importance of each component. Initially, we study the implementation of the idea presented in section 3.2 where

we project the input data to two constant vectors. Here one of the vectors is $\mu_U = \underline{0}$, where $\underline{0}$ is the all-zero vector, and the other vector is $\mu_P = \underline{a}$ where $a$ is a scalar that takes value from the set $\{1, 5, 50, 100\}$ (we report mean and standard deviation of accuracy of all trials), and $\underline{a}$ is the vector of all $a$'s. This experiment is referred to as "Constant encodings" in Table 3.

Next, we implement the idea while considering two normal distributions $U \sim \mathcal{N}(\underline{0}, I)$ and $P \sim \mathcal{N}(\underline{a}, I)$ instead of $\mu_U$, and $\mu_P$ respectively, where $\mathcal{N}(\underline{a}, I)$ is the normal distribution that has a mean vector of all $a$'s, and identity covariance matrix. Again, we let $a$ to take values from the set $\{1, 5, 50, 100\}$ (we report mean and standard deviation of accuracy of all trials). In this case, we penalize the KL divergence between the encoded vectors and the two normal distributions $U$ and $P$. This variant is named "distributional encodings" in Table 3.

Thirdly, we assess the impact of the number of codebook vectors. We implement the proposed method using the minimal feasible number of codebook vectors, which is two vectors.The resulting accuracy is presented in Table 3 and referred to as "VQ (2 updated C.B. vectors).

Fourth, we explore the significance of updating the codebook vectors during the vector quantization of the representation space. We replicated our previous experiments but this time without any updates to the codebook vectors. In this case, because the idea of the method relies on having two distinct magnitudes of vectors in the representation space, we initialize the codebook vectors to have two modes, such that half of the codebook vectors are initialized from a normal distribution $\mathcal{N}_1(\underline{0}, I))$, and the other half is initialized from a normal distribution $\mathcal{N}_2(\underline{a}, I))$. Here $a$ took values from the set $\{1, 5, 50, 100\}$. We refer to this method in Table 3 as "VQ (No updates)".

Lastly, we revisited the prior configuration but with a restriction to just two codebook vectors. The objective was to project both positive and negative samples onto one of these two vectors. This technique is denoted as "VQ (2 fixed C.B. vectors)" in Table 3.

| Ablation Studies | | | | |
|---|---|---|---|---|
| **Method** | Accuracy% ($\pm$std) | | | |
| **Constant Encodings** | 73($\pm$4) | | | |
| **Distributional Encodings** | 69($\pm$7) | | | |
| **VQ (2 updated C.B. vectors)** | 97($\pm$0.5) | | | |
| | $\mathcal{N}(\underline{0}, I), \mathcal{N}(\underline{1}, I)$ | $\mathcal{N}(\underline{0}, I), \mathcal{N}(\underline{5}, I)$ | $\mathcal{N}(\underline{0}, I), \mathcal{N}(\underline{50}, I)$ | $\mathcal{N}(\underline{0}, I)), \mathcal{N}(\underline{100}, I)$ |
| **VQ (No updates)** | 78.9($\pm$4.3) | 95.1($\pm$0.7) | 96.7($\pm$1) | 88.9($\pm$3.2) |
| **VQ (2 fixed C.B. vectors)** | 76.3($\pm$3) | 97.3($\pm$0.3) | 96.2($\pm$0.8) | 91.4($\pm$2.7) |

Table 3: Ablation studies conducted on AFHQ dataset

The conducted experiments and ablation studies show the efficiency of learning a new representation to learn from PU data, and the significance introduced by the quantization of the representation space. Although the idea of learning a new representation space stems from the simple mathematical steps shown in section 3.2, the conducted ablation studies show that quantization helps achieve a clear separation between the positive and negative data in the unlabeled set. The ablation studies also demonstrate the importance of allowing update of the codebook vectors when adopting a vector-quantized representation space. Since the choice of the means of the codebook vectors seems to impact the performance (as evident in the last two rows in Table 3), initializing all codebook vectors with zero mean and updating them during training eliminates the need to fine-tune the means at initialization.

# 6   CONCLUSION

This work addresses the PU learning problem using a simple and yet effective method based on applying the K-means clustering algorithm in a learned representation space. The main idea of this paper comes from the simplicity of the PU learning problem in low-dimensional settings, as Figure 1 illustrates. The failure of some existing PU learning methods on typical PU learning problems is inherent to the high-dimensional complexities of the data. The primary objective of this paper is to address the limitations of existing techniques when used on high-dimensional data by learning a new representation space for the data such that it imitates the phenomenon observed in low-dimensional settings. The learning process of the representation space is optimized to produce two distinct and separable clusters representing the positive and negative class distributions. Quantizing the learned representation space is shown to improve the performance of the method in producing separable clusters. Comparison of the proposed method to current state-of-the-art PU learning techniques shows that our method outperforms others in terms of accuracy across 4 different imaging datasets.

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

# A APPENDIX

In this section we wish to justify the informal theorem. We investigate the minimization function mentioned in equation 5. We note that this loss can be rewritten in the form

$$\hat{\mathcal{L}}(\theta) := \|f(\theta) - y\|_2^2$$

where

$$
f(\theta) :=
\begin{bmatrix}
\frac{1}{\sqrt{n_p}} v(x_1^p; \theta) \\
\frac{1}{\sqrt{n_p}} v(x_2^p; \theta) \\
\vdots \\
\frac{1}{\sqrt{n_p}} v(x_{n_p}^p; \theta) \\
\frac{\sqrt{\alpha}}{\sqrt{n_{up}}} v(x_1^{up}; \theta) \\
\frac{\sqrt{\alpha}}{\sqrt{n_{up}}} v(x_2^{up}; \theta) \\
\vdots \\
\frac{\sqrt{\alpha}}{\sqrt{n_{up}}} v(x_{n_{up}}^{up}; \theta) \\
\frac{\sqrt{1-\alpha}}{\sqrt{n_{un}}} v(x_1^{un}; \theta) \\
\frac{\sqrt{1-\alpha}}{\sqrt{n_{un}}} v(x_2^{un}; \theta) \\
\vdots \\
\frac{\sqrt{1-\alpha}}{\sqrt{n_{un}}} v(x_{n_{un}}^{un}; \theta)
\end{bmatrix}
\quad \text{and} \quad
y :=
\begin{bmatrix}
\frac{1}{\sqrt{n_p}} \mathbf{1}_{n_p} \otimes \mu_P \\
\frac{\sqrt{\alpha}}{\sqrt{n_{up}}} \mathbf{1}_{n_{up}} \otimes \mu_U \\
\frac{\sqrt{1-\alpha}}{\sqrt{n_{un}}} \mathbf{1}_{n_{un}} \otimes \mu_U
\end{bmatrix}
$$

where $\otimes$ denotes the Kronecker product. With this nonlinear least squares formulation one can use well established Neural Tangent Kernel (NTK) theory to show that the for sufficiently wide networks and sufficiently large scale of initialization the iterative updates and the output of the network remain close to that of the iterative updates on a linear problem of the form

$$\|J\theta - y\|_2^2 \tag{7}$$

where $J$ denotes the Jacobian of the mapping $f$ at random initialization. This is a direction consequence of the argument in Section 5.3 of (Oymak et al., 2019) combined with NTK eigenvalue characterizations for deep convolutional networks in (Du et al., 2019). This argument is by now standard, and thus we omit unnecessary repetition given the informal/qualitative statement of our theorem and focus on the linearized form in equation 7. Without loss of generality we can focus on the case where $\mu_P$ and $\mu_U$ are scalar valued as the argument in the general case follows the exact same proof and can be thought of as repeating the scalar argument across the coordinates of $\mu_P/\mu_U$. The loss in this case can also be alternatively written in the form

$$\tilde{\mathcal{L}}(\theta) = \min_\theta \frac{1}{np} \|J_p\theta - \mu_P \mathbf{1}\|^2 + \frac{\alpha}{n_{up}} \|J_{up}\theta - \mu_U \mathbf{1}\|^2 + \frac{1-\alpha}{n_{un}} \|J_{up}\theta - \mu_{un} \mathbf{1}\|^2 \tag{8}$$

Where $J_p \in \mathbb{R}^{n_p \times d}$ is the Jacobian matrix corresponding to the positive labeled samples, similarly, the matrices $J_{up} \in \mathbb{R}^{n_{up} \times d}$ and $J_{un} \in \mathbb{R}^{n_{un} \times d}$ correspond to the unlabeled positive and negative samples, respectively, and $\theta \in \mathbb{R}^{d \times 1}$ is the linear model, and $\mathbf{1}$ is the all 1 vector.

For convenience, the three loss terms can be combined into a tall concatenated matrix as follows:

$$
\tilde{\mathcal{L}}(\theta) =
\left\|
\begin{bmatrix}
\frac{J_p}{\sqrt{n_p}}\theta - \frac{\mu_P}{\sqrt{n_p}}\mathbf{1} \\
\frac{\sqrt{\alpha}}{\sqrt{n_{up}}} J_{up}\theta - \frac{\sqrt{\alpha}\mu_U}{\sqrt{n_{up}}}\mathbf{1} \\
\frac{\sqrt{1-\alpha}}{\sqrt{n_{un}}} J_{un}\theta - \frac{\sqrt{1-\alpha}\mu_U}{\sqrt{n_{un}}}\mathbf{1}
\end{bmatrix}
\right\|^2
\tag{9}
$$

Thus in this case $J$ corresponds to $\begin{bmatrix} \frac{J_p}{\sqrt{n_p}} \\ \frac{\sqrt{\alpha}}{\sqrt{n_{up}}} J_{up} \\ \frac{\sqrt{1-\alpha}}{\sqrt{n_{un}}} J_{un} \end{bmatrix}$ and $y$ to $\begin{bmatrix} \frac{\mu_P}{\sqrt{n_p}}\mathbf{1} \\ \frac{\sqrt{\alpha}\mu_U}{\sqrt{n_{up}}}\mathbf{1} \\ \frac{\sqrt{1-\alpha}\mu_U}{\sqrt{n_{un}}}\mathbf{1} \end{bmatrix}$.

Applying gradient descent to minimize the loss function, the update rule for $\theta$ is :

$$\theta_{t+1} = \theta_t - \eta J^T(J\theta_t - y)$$

Where $\eta$ is the learning rate. Defining the residual vector $r_t := J\theta_t - y$ after $t$ iterations we have

$$
\begin{aligned}
r_t = J\theta_t - y &= J\theta_{t-1} - y - \eta J J^T(J\theta_{t-1} - y)\\
&= (I - \eta J J^T)(J\theta_{t-1} - y)\\
&= (I - \eta J J^T)r_{t-1}\\
&= \left(I - \eta J J^T\right)^t r_0
\end{aligned}
\tag{10}
$$

With sufficiently small or asymmetric initialization ((Oymak et al., 2019)) we can ensure $\theta_0 \approx 0$ which implies that the initial residual is $r_0 = J\theta_0 - y \approx -y$, hence,

$$J\theta_t = y - \left(I - \eta J J^T\right)^t y \tag{11}$$

Now consider the vector $\mathbf{w} = \begin{bmatrix} \frac{\sqrt{\alpha}\mathbf{1}}{\sqrt{n_p}} \\ \frac{-\mathbf{1}}{\sqrt{n_{up}}} \\ 0 \end{bmatrix}$ . The critical observation is that this vector is approximately in the null space of $J^T$. To see this note that

$$J^T w = \left(\frac{J_p}{\sqrt{n_p}}\right)^T \frac{\sqrt{\alpha}\mathbf{1}}{\sqrt{n_p}} - \left(\frac{\sqrt{\alpha}}{\sqrt{n_{up}}}J_{up}\right)^T \frac{\mathbf{1}}{\sqrt{n_{up}}} = \sqrt{\alpha}\left(\frac{J_p^T\mathbf{1}}{n_p} - \frac{J_{up}^T\mathbf{1}}{n_{up}}\right) \tag{12}$$

$\frac{J_p^T\mathbf{1}}{n_p}$ and $\frac{J_{up}^T\mathbf{1}}{n_{up}}$ are simply the empirical average of the NTK features over the labeled and unlabeled positive pairs. Since these two distributions are identical they converge to the same population mean. Let us denote this common mean by $\phi$. Thus,

$$\|J^T w\| = \sqrt{\alpha}\left\|\frac{J_p^T\mathbf{1}}{n_p} - \phi - \left(\frac{J_{up}^T\mathbf{1}}{n_{up}} - \phi\right)\right\| \leq \sqrt{\alpha}\left\|\frac{J_p^T\mathbf{1}}{n_p} - \phi\right\| + \sqrt{\alpha}\left\|\frac{J_{up}^T\mathbf{1}}{n_{up}} - \phi\right\| \leq \sqrt{\alpha}\sqrt{\delta}$$

where the latter holds with high probability do to the concentration of the empirical mean around the true mean under mild technical assumptions about the NTK kernel and data distributions.Indeed, if the features are sub-Gaussian (e.g. bounded) one can show that $\sqrt{\delta}$ scales with $\max\left(1/\sqrt{n_p}, 1/sqrtn_{up}\right)$ and can thus be made arbitrarily small for a sufficiently large data set. To continue define the unit norm vector $\hat{\mathbf{w}} = \frac{\mathbf{w}}{\sqrt{1+\alpha}}$ and note that

$$\hat{w}^T J J^T \hat{w} \leq \frac{\alpha}{\alpha+1}\delta \leq \delta.$$

Now, we can decompose $y$ into it's orthogonal projections onto $\hat{\mathbf{w}}$ where $\hat{\mathbf{w}} = \frac{\mathbf{w}}{\sqrt{1+\alpha}}$: $y = y_\| + y_\perp = \hat{\mathbf{w}}\hat{\mathbf{w}}^T y + \left(I - \hat{\mathbf{w}}\hat{\mathbf{w}}^T\right)y.$

To continue note that

$$
\begin{aligned}
y_\perp := \left(I - \hat{\mathbf{w}}\hat{\mathbf{w}}^T\right) y &= y - \mathbf{w}\frac{\mathbf{w}^T y}{1 + \alpha} \\
&= y - \mathbf{w}\frac{\sqrt{\alpha}\mu_P - \sqrt{\alpha}\mu_U}{1 + \alpha} \\
&= y - \begin{bmatrix} \frac{\alpha}{1+\alpha}\frac{1}{\sqrt{n_p}}(\mu_P - \mu_U) \\ -\frac{\sqrt{\alpha}}{1+\alpha}\frac{1}{\sqrt{n_{up}}}(\mu_P - \mu_U) \\ \mathbf{0} \end{bmatrix} \\
&= \begin{bmatrix} \frac{\mu_P}{\sqrt{n_p}}\mathbf{1} \\ \frac{\sqrt{\alpha}\mu_U}{\sqrt{n_{up}}}\mathbf{1} \\ \frac{\sqrt{1-\alpha}\mu_U}{\sqrt{n_{un}}}\mathbf{1} \end{bmatrix} - \begin{bmatrix} \frac{\alpha}{1+\alpha}\frac{1}{\sqrt{n_p}}(\mu_P - \mu_U) \\ -\frac{\sqrt{\alpha}}{1+\alpha}\frac{1}{\sqrt{n_{up}}}(\mu_P - \mu_U) \\ \mathbf{0} \end{bmatrix} \\
&= \begin{bmatrix} \frac{1}{\sqrt{n_p}}\left[\left(1 - \frac{\alpha}{1+\alpha}\right)\mu_P + \frac{\mu_U}{1+\alpha}\right]\mathbf{1} \\ \frac{\sqrt{\alpha}}{\sqrt{n_{up}}}\left[\mu_P + \left(1 - \frac{1}{1+\alpha}\right)\mu_U\right]\mathbf{1} \\ \frac{\sqrt{1-\alpha}\mu_U}{\sqrt{n_{un}}}\mathbf{1} \end{bmatrix} \\
&= \begin{bmatrix} \frac{1}{\sqrt{n_p}}\frac{\mu_P+\alpha\mu_U}{1+\alpha}\mathbf{1} \\ \frac{\sqrt{\alpha}}{\sqrt{n_{up}}}\frac{\mu_P+\alpha\mu_U}{1+\alpha}\mathbf{1} \\ \frac{\sqrt{1-\alpha}\mu_U}{\sqrt{n_{un}}}\mathbf{1} \end{bmatrix}
\end{aligned}
$$

Furthermore,

$$
J^T y_\perp = \frac{1}{n_p}\frac{\mu_P + \alpha\mu_U}{1 + \alpha}J_p^T\mathbf{1} + \frac{\alpha}{n_{up}}\frac{\mu_P + \alpha\mu_U}{1 + \alpha}J_{up}^T\mathbf{1} + \frac{1 - \alpha}{n_{un}}\mu_U J_{un}^T\mathbf{1}
$$

Now using concentration of the rows of different $J$ the above is approximately equal to the following with high probability

$$
J^T y_\perp \approx \left(\mu_P + \alpha\mu_U\right)\phi + (1 - \alpha)\mu_U\tilde{\phi}
$$

where $\phi$ and $\tilde{\phi}$ are the average of the NTK features in the positive and unlabeled negative data. Thus, for $\hat{v} = y_\perp/\|y_\perp\|_2$ we have

$$
v^T J J^T v = \frac{1}{\|y_\perp\|_2^2}y_\perp^T J J^T y_\perp \geq (1 + \alpha)\frac{\|\left(\mu_P + \alpha\mu_U\right)\phi + (1 - \alpha)\mu_U\tilde{\phi}\|_2^2}{\mu_P^2 + \mu_U^2 + 2\alpha\mu_P\mu_U} := \Delta
$$

Thus in the direction of $\hat{w}$ the NTK kernel $J J^T$ is small where as in the direction $\hat{v}$ it is large. Intuitively, this implies that $(I - \eta J J^T)^t y_\perp$ is small for a sufficiently large $t$ where as $(I - \eta J J^T)^t y_\| \approx y_\|$. Indeed, we can make this intuition precise and prove that

$$
\|(I - \eta J J^T)^t y_\perp\|_2 \leq (1 - \eta\Delta)^t\|y_\perp\|_2 \quad \text{and} \quad \|y_\| - (I - \eta J J^T)^t y_\|\|_2 \leq \left(1 - (1 - \eta\delta)^t\right)\|y_\|\|_2
$$

Since $\delta$ can be made arbitrarily small for a sufficiently large data set we have $\delta << \Delta$ therefore for a broad range of values of $\eta$ one can find a stopping time $T$ where both terms are very small. For instance for $\eta = \frac{1}{2\Delta}$ picking any stopping time obeying

$$
\log\left(\frac{2}{\epsilon}\right) \leq T \leq \frac{\log\left(1 - \frac{\epsilon}{2}\right)}{\log\left(1 - \frac{\delta}{\Delta}\right)}
$$

we have

$$
\|(I - \eta J J^T)^T y_\perp\|_2 \leq \frac{\epsilon}{2}\|y_\perp\|_2 \quad \text{and} \quad \|y_\| - (I - \eta J J^T)^T y_\|\|_2 \leq \frac{\epsilon}{2}\|y_\|\|_2
$$

using the above identities we conclude that

$$
\|J\theta_T - y_\perp\| = \|y_\| - \left(I - \eta J J^T\right)^T y_\| - \left(I - \eta J J^T\right)^T y_\perp\| \leq \epsilon\|y\|
$$

This formally proves that for an appropriate stopping time $T$

$$J\theta_T \approx y_\perp \tag{13}$$

Pulling back the definition of $J$:

$$J\theta_T = \begin{bmatrix} \frac{J_p}{\sqrt{n_p}}\theta_T \\ \frac{\sqrt{\alpha}}{\sqrt{n_{up}}}J_{up}\theta_T \\ \frac{\sqrt{1-\alpha}}{\sqrt{n_{un}}}J_{un}\theta_T \end{bmatrix} = \begin{bmatrix} \frac{1}{\sqrt{n_p}}\frac{\mu_P+\alpha\mu_U}{1+\alpha}\mathbf{1} \\ \frac{\sqrt{\alpha}}{\sqrt{n_{up}}}\frac{\mu_P+\alpha\mu_U}{1+\alpha}\mathbf{1} \\ \frac{\sqrt{1-\alpha}\mu_U}{\sqrt{n_{un}}}\mathbf{1} \end{bmatrix} \tag{14}$$

Resulting in:

$$J_p\theta_T = J_{up}\theta_T \approx \frac{\mu_P + \alpha\mu_U}{1+\alpha}\mathbf{1}, \quad \text{and} \quad J_{un}\theta_t \approx \mu_U\mathbf{1} \quad \square$$

# B  APPENDIX

## B.1  TESTING THE SPEED OF K-MEANS ALGORITHM AFTER EACH ITERATION

We test the speed of running the K-means algorithm when training on different datasets, and Table 4 shows the results. The table shows the average and standard deviation of the time elapsed by the K-means algorithm after each training epoch in seconds for each of three datasets, along with the dimensions of the learned representation the algorithm is run on and the number of samples. Although increasing data size will increase the time of running the algorithm, we expect the time to only increase linearly as the time complexity of K-means is linear in dimensions and number of samples. It should be pointed out that the K-means algorithm in the proposed method is run on the learned representation space, which is in lower dimensions than the input data.

| Dataset | MNIST | Fashion MNIST | AFHQ |
|---|---|---|---|
| # of Samples, # of dimensions | $19000, 49$ | $19000, 49$ | $3300, 1024$ |
| K-means Elapsed Time (s) | $0.0207 \pm 0.0038$ | $0.0292 \pm 0.0174$ | $0.3681 \pm 0.0071$ |

Table 4: Time Elapsed in running K-means algorithm after each training epoch

## B.2  TESTING DIFFERENT $\alpha$ VALUES

We conducted an experiment to assess the effectiveness of our method when the likelihood of positive samples was either less or greater than that of negative samples, characterized by different $\alpha$ values. We then compared the outcomes of this experiment with the results obtained using $TED^n$ method in Table 5.

| | $\alpha = 0.2$ | $\alpha = 0.4$ | $\alpha = 0.5$ | $\alpha = 0.6$ | $\alpha = 0.8$ |
|---|---|---|---|---|---|
| $TED^n$ | $86.9 \pm 1$ | $89.5 \pm 2.5$ | $89.9 \pm 14.9$ | $81.5 \pm 3.7$ | $71.1 \pm 2$ |
| VQ K-means | $97.6 \pm 1.1$ | $96.5 \pm 0.7$ | $95.4 \pm 1.3$ | $84.3 \pm 1.1$ | $70.2 \pm 2.1$ |

Table 5: Classification results with different $\alpha$ values

## B.3  COMPARISON AGAINST MORE METHODS

Here we provide comparisons of the proposed method to Robust-PU (Zhu et al., 2023) and Dist-PU (Zhao et al., 2022). Table 6 shows the results table including the two additional methods.

| | D-GAN | $TED^n$ | | $Observer$ | | VQ K-means | Robust-PU | Dist-PU |
|---|---|---|---|---|---|---|---|---|
| | **Early Stop** | **50** | **100** | **50** | **100** | **highest distance** | | |
| **AFHQ (Cats vs. Dogs)** | $50.3 \pm 0.2$ | $86.8 \pm 12$ | $89.9 \pm 14.9$ | $91 \pm 1.1$ | $90.1 \pm 3.2$ | $95.3 \pm 1.3$ | $65.8 \pm 1.4$ | $83.9 \pm 0.6$ |
| **CIFAR (Animal vs. Not Animal)** | $82 \pm 1.1$ | $88 \pm 2.5$ | $87.7 \pm 4.6$ | $89.6 \pm 0.7$ | $88.8 \pm 1.7$ | $91.1 \pm 0.7$ | $83.8 \pm 1.3$ | $87 \pm 1.1$ |
| **MNIST (Even vs. Odd)** | $98.3 \pm 0.1$ | $97.7 \pm 0.4$ | $97.7 \pm 0.4$ | $98.3 \pm 0.2$ | $97.8 \pm 1.6$ | $98.1 \pm 0.1$ | $97.1 \pm 0.4$ | $91.4 \pm 0.5$ |
| **Binarized Fashion MNIST** | $89.6 \pm 0.2$ | $88.5 \pm 0.9$ | $88.1 \pm 1$ | $92.6 \pm 0.3$ | $92 \pm 1$ | $93.3 \pm 0.75$ | $90.7 \pm 0.2$ | $86 \pm 1.6$ |

Table 6: Summary of experimental results averaged over 5 trials

