# OpenReview forum: "Learning A Disentangling Representation For PU Learning"
_ICLR.cc/2024/Conference — ICLR 2024 Conference Withdrawn Submission_

### Official Review · Reviewer_ShRs · 2023-10-17

**Soundness:** 3 good
**Presentation:** 2 fair
**Contribution:** 2 fair
**Rating:** 3
**Confidence:** 4

**Summary:**

This paper proposes a new PU learning method. Specifically, the authors develop a loss function that can be used to project the unlabeled data into two (positive and negative) clusters that can be easily identified. They adopt a vector quantization technique for the learned representations to amplify the separation between the learned unlabeled data clusters.

**Strengths:**

1.	The studied problem in this paper is very important.
2.	The experiments are sufficient.

**Weaknesses:**

1.	The authors claim that the existing PU learning methods will suffer a gradual decline in performance as the dimensionality of the data increases. It would be better if the authors can visualize this effect. This is very important as this is the research motivation of this paper.
2.	Since the authors claim that the high dimensionality is harmful for the PU methods, have the authors tried to firstly implement dimension reduction via some existing approaches and then deploy traditional PU classifiers?
3.	In problem setup, the authors should clarify whether their method belongs to case-control PU learning or censoring PU learning, as their generation ways of P data and U data are quite different.
4.	The proposed algorithm contains Kmeans operation. Note that if there are many examples with high dimension, Kmeans will be very inefficient.
5.	The authors should compare their algorithm with SOTA methods and typical methods on these benchmark datasets.
6.	The figures in this paper are in low quality. Besides, the writing of this paper is also far from perfect.

**Questions:**

see the weakness part.

---

> ### Author Response · Authors · 2023-11-21
>
> We appreciate your insightful comments and recommendations. Here we try to address the raised points:
>
> > The authors claim that the existing PU learning methods will suffer a gradual decline in performance as the dimensionality of the data increases. It would be better if the authors can visualize this effect. This is very important as this is the research motivation of this paper.
>
> We argue that there is a pattern of decrease in performance in the experiments table, as all the methods perform better on MNIST and Fashion MNIST compared to their performance on CIFAR and AFHQ datasets that are of higher dimensions. More notably, looking at the performance of the D-GAN method, it is evident that the performance of the method completely breaks down on AFHQ dataset which has the highest dimensions. It is relevant to mention that the results show that reducing the dimensionality of the data using the proposed method is very effective in making learning easier. We compare different methods applied directly to the input data to the simple K-means algorithm when applied to the proposed learned representation to directly show that learning the new lower-dimensional representation is an effective approach. We have to emphasize that the proposed learned representation can be used as input to any of the other methods. We argue that since a simple K-means algorithm applied to the learned representation is shown to perform well, applying SOTA methods to the learned representation can be even more effective.
>
> > Since the authors claim that the high dimensionality is harmful for the PU methods, have the authors tried to firstly implement dimension reduction
>
> Indeed, one of the main innovations in the presented work is to first apply a dimensionality reduction method to the data before learning the binary classifier from it. Figure 2 in the paper shows that the proposed learned representation is more meaningful in the context of PU learning (as it separates unlabeled positive from unlabeled negative samples) than one of the SOTA dimensionality reduction methods VQ-VAE.
>
> > In problem setup, the authors should clarify whether their method belongs to case-control PU learning or censoring PU learning
>
>
> We appreciate the reviewer's note about the need to clearly mention the kind of setup we are dealing with. The setup dealt with in this paper is the case-control PU setup, in which the labeled samples are Selected Completely At Random (SCAR) from the conditional distribution $p(x|y=1)$, and the unlabeled data is sampled from $p(x)$.
>
>
> > The proposed algorithm contains Kmeans operation. Note that if there are many examples with high dimension, Kmeans will be very inefficient.
>
> We tested the speed of the K-means algorithm when training on different datasets and included the results in Table 4 in section B.1 in the appendix. We expect the numbers shown in the table to increase only linearly as the number of samples or dimension increase, as the time complexity of the algorithm is linear in dimension and number of samples. It should be pointed out that the Kmeans algorithm in the proposed method is run on the learned representation space, which is in lower dimensions than the input data.
>
>
> > The authors should compare their algorithm with SOTA methods
>
> Thank you for the suggestion. We included two more methods, Robust-PU and Dist-PU in a comparison against the proposed method, and added the results to Table 6 in section B.3 in the appendix. We are working on adding more comparisons to the appendix section in the final version.

---

> > ### Author Response · Authors · 2023-11-21
> > **Tables**
> >
> > Here are the tables we added to the appendix and referred to in the comment:
> >
> > Testing the speed of running the K-means algorithm after each epoch (in seconds):
> >
> > | **Dataset**                          | **MNIST**            | **Fashion MNIST**    | **AFHQ**            |
> > |--------------------------------------|----------------------|----------------------|---------------------|
> > | **# of Samples, # of dimensions**    | $19000, 49$          | $19000, 49$          | $3300, 1024$        |
> > | **K-means Elapsed Time (s)**         | $0.0207 \pm 0.0038$  | $0.0292 \pm 0.0174$  | $0.3681 \pm 0.0071$ |
> >
> > Performance comparison table including the two added methods:
> >
> > |                                    | **D-GAN (Early Stop)** | **TED^n (100)** | **Observer (100)** | **VQ K-means (highest distance)** | **Robust-PU** | **Dist-PU** |
> > |------------------------------------|------------------------|-----------------|--------------------|-----------------------------------|---------------|-------------|
> > | **AFHQ (Cats vs. Dogs)**           | $50.3 \pm 0.2$         | $89.9\pm 14.9$  | $90.1\pm 3.2$      | $95.3\pm 1.3$                     | $65.8\pm 1.4$ | $83.9\pm 0.6$|
> > | **CIFAR (Animal vs. Not Animal)**  | $82 \pm 1.1$           | $87.7\pm 4.6$   | $88.8\pm 1.7$      | $91.1\pm 0.7$                     | $83.8\pm 1.3$ | $87\pm 1.1$  |
> > | **MNIST (Even vs. Odd)**           | $98.3 \pm 0.1$         | $97.7\pm 0.4$   | $97.8\pm 1.6$      | $98.1\pm 0.1$                     | $97.1\pm 0.4$ | $91.4\pm 0.5$|
> > | **Binarized Fashion MNIST**        | $89.6 \pm 0.2$         | $88.1\pm 1$     | $92\pm 1$          | $93.3\pm 0.75$                    | $90.7\pm 0.2$ | $86\pm 1.6$  |
> >
> > Appendix B contains additional notes for your reference.

---

### Official Review · Reviewer_w13V · 2023-10-19

**Soundness:** 1 poor
**Presentation:** 3 good
**Contribution:** 2 fair
**Rating:** 3
**Confidence:** 3

**Summary:**

This paper works on PU learning and proposes a new representation learning method for it. The paper uses a codebook to store representations and forces P and U data to be similar to different codebook vectors, respectively. Then, they use a K-means algorithm to cluster feature representations and derive the classifier. Experiments validate the effectiveness of the proposed approach.

**Strengths:**

- The paper is well written.
- The idea of introducing codebook representations into PU learning is novel.

**Weaknesses:**

- It is still unclear to me why the proposed method works for PU learning. Although the authors provided some theoretical explanations, I am still not clear why the proposed method can separate feature representations of P and N data.
- The proposed method is influenced by the center representations of P and U data ($\mu_P$ and $\mu_U$). If the two representations are too close, there seems to be no guarantee that the method will work well.
- In Eq.(6), the authors claim that they do not need $\alpha$. However, they still need to know the labels of the unlabeled data. But if we know the labels of the unlabeled data, we can calculate $\alpha$. So I do not think the analysis is useful here.
- The experiment design is too simple. The authors should include more experiments, such as more compared approaches, and more experimental settings (such as different $\alpha$). The current experiments are too simple to validate the effectiveness of the proposed approach.

**Questions:**

- Why does the proposed method work well?
- Is the method affected by the feature separability of the training data?
- Can the authors add more experiments to verify the proposal?

---

> ### Author Response · Authors · 2023-11-21
>
> Thank you the valuable feedback and suggestions.
>
> > Why does the proposed method work well?
>
> Section 3.2 gives a mathematical explanation of why the proposed method should work, and the provided theorem in the same section articulates that. The method's success in separating the unlabeled data into positive and negative clusters can be simply explained as follows: The proposed loss function in the paper projects the unlabeled samples into a vector $\mu_{U}$ and the positive samples into a vector $\mu_P$. Considering only the positive samples (that have the same distribution) in both the labeled and unlabeled sets, they are projected to two different vectors, which has to be relatively more difficult than projecting the negative samples in the unlabeled data to $\mu_{U}$. The proposed method relies on the difficulty of projecting some samples that have the same distribution to two different vectors. This difficulty results in the encoder network projecting positive samples to a linear combination of the two vectors $\mu_{U}$ and $\mu_{P}$, while more easily projecting the negative samples to $\mu_{U}$. Hence, the sought-after separation is achieved.
>
> > The proposed method is influenced by the center representations of P and U data ( and ). If the two representations are too close, there seems to be no guarantee that the method will work well.
>
> We agree that the choice of $\mu_{U}$ and $\mu_{P}$ has an effect on the performance of the method. The ablation study section emphasizes that by making various experiments with these choices and concludes that the most reliable way to choose $\mu_{U}$ and $\mu_{P}$ is to learn them during training, which is part of the proposed loss function in Eq. (1) in the paper. The loss function in Eq. (1) forces the codebook vectors to align with the output of the encoder, which is forced to project positive data to the vector with the highest magnitude and the unlabeled data to the vector of the lowest magnitude. The distance between $\mu_{U}$ and $\mu_{P}$ is reflected in the distance plot shown in Figure 4 in the paper.
>
> > In Eq.(6), the authors claim that they do not need $\alpha$. However, they still need to know the labels of the unlabeled data.
>
> Please notice that the last two terms in Eq (6) are just a summation over all unlabeled data, and we don't need the label of the samples to do this summation. In Eq. (6) Both of the unlabeled positive and unlabeled negative samples are projected to $\mu_{U}$. Hence, there is no need to know which of the unlabeled samples are positive and which are negative.
>
> > The experiment design is too simple. The authors should include more experiments, such as more compared approaches, and more experimental settings (such as different $\alpha$ values )
>
> Thank you for the suggestion. We included two more methods, Robust-PU and Dist-PU, in a comparison against the proposed method, and added the results in Table 6 in appendix section B.3. Also, we have conducted an experiment where we vary the value of $\alpha$ and compare the proposed method to $TED^{n}$ method. Given the limited time available for the review period and the multiple methods and data configurations to be tested, we were unable to include an extensive table covering all methods and configurations for now. We are currently compiling these results and plan to include them in the appendix of the paper in the final version.
>
> > Is the method affected by the feature separability of the training data?
>
> Yes, the similarity between the features of the positive and negative samples in the unlabeled set is believed to affect the performance of the proposed method as it will make it more difficult for the encoder to encode them to sufficiently different representations. But one should note that this is the case for any other method. The feature separability of the training data is influential in any classification problem in general.

---

> > ### Comment · Reviewer_w13V · 2023-11-21
> > **Thanks for the reply**
> >
> > Thanks for the additional experiments!
> >
> > For the first reply, I am still not convinced. Are there any evidence showing that the **difficulty** of P and U data for projection are different? Adding theoretical or empirical results may prove it.
> >
> > The second reply seems not addressing my question. I mean if P and U data are very similar (e.g. the proportion of P data is very large in U), the method may not work because it only depends on the two representation centroids of P and U data.
> >
> > For the third reply, my question is addressed. However, why not just writing U data? Writing them separately may show that their labels are identified.

---

> > > ### Author Response · Authors · 2023-11-21
> > > **Thanks for the response**
> > >
> > > > For the first reply, I am still not convinced. Are there any evidence showing that the difficulty of P and U data for projection are different? Adding theoretical or empirical results may prove it.
> > >
> > > Theorem 1 in page 5 of the paper states that training a network to project unlabeled data to a vector $\mu_{U}$ and the positive data to a vector $\mu_{P}$ will result in projecting the positive labeled and positive unlabeled samples to a linear combination of $\mu_{U}$ and $\mu_{P}$, and the negative samples in the unlabeled data to $\mu_{U}$, hence, the sougth-after separation is achieved. Please refer to appendix A for a detailed justification and proof of the theorem. The results shown in the paper is also supporting the validity of the approach, as the ultimate binary classification is attained with high accuracies, which can only happen if the separation of positive and negative samples is achieved.
> > >
> > > > The second reply seems not addressing my question. I mean if P and U data are very similar (e.g. the proportion of P data is very large in U), the method may not work because it only depends on the two representation centroids of P and U data.
> > >
> > > Sorry for the misunderstanding. You are correct, if the unlabeled data consists of mostly positive samples, the whole training data will contain a relatively small number of negative samples, which is believed to affect the training of any PU learning method. We added an experiment in section B.2 in the appendix to test how the training changes as the proportion of the positive samples in the unlabeled data varies and compare the results to one of the competitive SOTA methods. We are working on including the rest of the methods to the table in the final version.
> > >
> > > > For the third reply, my question is addressed. However, why not just writing U data? Writing them separately may show that their labels are identified.
> > >
> > > We agree that writing the summation over the whole unlabeled data will make that clearer. Having it in the current form was just following the sequence of the preceding equations. We have added the summation over the unlabeled data to the current version of the paper for clarity. Thanks for the suggestion.

---

> > > > ### Author Response · Authors · 2023-11-21
> > > > **Tables**
> > > >
> > > > Here are the tables we added to the appendix and referred to:
> > > >
> > > > Comparisons table including the two added methods:
> > > >
> > > > |                                    | **D-GAN (Early Stop)** | **TED^n (100)** | **Observer (100)** | **VQ K-means (highest distance)** | **Robust-PU** | **Dist-PU** |
> > > > |------------------------------------|------------------------|-----------------|--------------------|-----------------------------------|---------------|-------------|
> > > > | **AFHQ (Cats vs. Dogs)**           | $50.3 \pm 0.2$         | $89.9\pm 14.9$  | $90.1\pm 3.2$      | $95.3\pm 1.3$                     | $65.8\pm 1.4$ | $83.9\pm 0.6$|
> > > > | **CIFAR (Animal vs. Not Animal)**  | $82 \pm 1.1$           | $87.7\pm 4.6$   | $88.8\pm 1.7$      | $91.1\pm 0.7$                     | $83.8\pm 1.3$ | $87\pm 1.1$  |
> > > > | **MNIST (Even vs. Odd)**           | $98.3 \pm 0.1$         | $97.7\pm 0.4$   | $97.8\pm 1.6$      | $98.1\pm 0.1$                     | $97.1\pm 0.4$ | $91.4\pm 0.5$|
> > > > | **Binarized Fashion MNIST**        | $89.6 \pm 0.2$         | $88.1\pm 1$     | $92\pm 1$          | $93.3\pm 0.75$                    | $90.7\pm 0.2$ | $86\pm 1.6$  |
> > > >
> > > > Comparing the performance of the method to $TED^{n}$ for different $\alpha$ values:
> > > >
> > > > |                     | **α = 0.2**        | **α = 0.4**        | **α = 0.5**         | **α = 0.6**        | **α = 0.8**        |
> > > > |---------------------|--------------------|--------------------|---------------------|--------------------|--------------------|
> > > > | **TED^n**           | $86.9 \pm 1$       | $89.5 \pm 2.5$     | $89.9 \pm 14.9$     | $81.5 \pm 3.7$     | $71.1 \pm 2$       |
> > > > | **VQ K-means**      | $97.6 \pm 1.1$     | $96.5 \pm 0.7$     | $95.4 \pm 1.3$      | $84.3 \pm 1.1$     | $70.2 \pm 2.1$     |
> > > >
> > > > Appendix B contains additional notes for your reference.

---

### Official Review · Reviewer_Wr1j · 2023-10-31

**Soundness:** 1 poor
**Presentation:** 2 fair
**Contribution:** 1 poor
**Rating:** 3
**Confidence:** 5

**Summary:**

The authors in this paper focus on positive-unlabeled (PU) learning and attempt to encode positive and unlabeled instances into a more discriminative representation space followed by a simple cluster method, such as K-means. They directly apply the existing vector quantization technique to project the unlabeled data into two distinct clusters. The experimental results show the effectiveness of the vector quantization method.

Though the idea of learning a disentangling representation for PU learning may be interesting, applying the existing vector quantization technique directly limits the contribution of this paper.

**Strengths:**

-	This paper is well-written and quite easy to follow.
-	The experimental results and ablation study show the effectiveness of the proposed method.

**Weaknesses:**

-	The innovation of this paper seems to be limited. In this paper, the authors directly employ the exited vector quantization technique [1] to learn a disentangling representation for PU learning with little modification. Though the idea of learning a disentangling representation for PU learning may be interesting, the contribution of this paper is very limited. Otherwise, there lacks of reference to the original paper “Neural discrete representation learning” [1] of the vector quantization technique.
-	There lack of some current PU approaches as baselines in experiments, such as Robust-PU [2], Dist-PU [3], P3Mix [4].
-	Equation (1) misses a “)”, and should be $sg(\mathbf{v}_j(\mathbf{x}_{i_p};\theta))$.

[1] Aaron Van Den Oord, and Oriol Vinyals. "Neural discrete representation learning." Advances in neural information processing systems 30 (2017).

[2] Zhangchi Zhu, Lu Wang, Pu Zhao, Chao Du, Wei Zhang, Hang Dong, Bo Qiao, Qingwei Lin, Saravan Rajmohan, and Dongmei Zhang. "Robust Positive-Unlabeled Learning via Noise Negative Sample Self-correction." In Proceedings of the 29th ACM SIGKDD Conference on Knowledge Discovery and Data Mining, pp. 3663-3673. 2023.

[3] Yunrui Zhao, Qianqian Xu, Yangbangyan Jiang, Peisong Wen, and Qingming Huang. 2022. Dist-PU: Positive-Unlabeled Learning From a Label Distribution Perspective. In Proceedings of the IEEE/CVF Conference on Computer Vision and Pattern Recognition. 14461–14470.

[4] Changchun Li, Ximing Li, Lei Feng, and Jihong Ouyang. 2022. Who is your right mixup partner in positive and unlabeled learning. In International Conference on Learning Representations.

**Questions:**

Please see the weakness for details.

---

> ### Author Response · Authors · 2023-11-21
>
> Thank you for the valuable feedback and suggestions.
>
> > The innovation of this paper seems to be limited. In this paper, the authors directly employ the exited vector quantization technique [1] to learn a disentangling representation for PU learning with little modification.
>
> Thank you for noticing the missing reference to the "Neural discrete representation learning" paper. We apologize and have now added the citation in the main text. Please note that both VQ-VAE and the proposed method are dimensionality reduction methods and both can be used as a first step in learning from PU data. Figure 2 in the paper addresses why the proposed method is superior to a direct adaptation of VQ-VAE in this context. The figure shows the difference between the learned representations of the proposed method vs. VQ-VAE. The figure clearly shows that a direct adaptation of VQ-VAE results in a learned representation that might be useful for image reconstruction but is obviously less meaningful compared to the proposed method for the ultimate goal of disentangling positive from negative samples in the unlabeled set. It is significant to mention that while the vector quantization technique is adapted from the original VQ-VAE paper, it is applied to a loss function that is different from the VAE loss function and in a different context to achieve the learned representation separability.
>
> > There lack of some current PU approaches as baselines in experiments
>
> Thanks for bringing these paper to our attention. We included the mentioned methods, Robust-PU and Dist-PU in a comparison against the proposed method, and added the results in Table 6 in section B.3 in the appendix. Due to the limited response time and the lack of readily available online implementations, we were unable to include the P3Mix method in the current table. However, we are in the process of implementing it and plan to add it to the comparisons in the final version of our work. It is relevant to mention that the papers Dist-PU and Robust-PU compare their methods against P3Mix in their papers and show that their methods outperform P3Mix.
>
> > Equation (1) misses a “)”, and should be $sg(\mathbf{v}j(\mathbf{x}{i_p};\theta))$.
>
> Thank you for catching this typo. It has now been edited in the paper.

---

> > ### Author Response · Authors · 2023-11-21
> > **Tables**
> >
> > Here are the table we added to appendix B.3 and referred to in the comment:
> >
> > |                                    | **D-GAN (Early Stop)** | **TED^n (100)** | **Observer (100)** | **VQ K-means (highest distance)** | **Robust-PU** | **Dist-PU** |
> > |------------------------------------|------------------------|-----------------|--------------------|-----------------------------------|---------------|-------------|
> > | **AFHQ (Cats vs. Dogs)**           | $50.3 \pm 0.2$         | $89.9\pm 14.9$  | $90.1\pm 3.2$      | $95.3\pm 1.3$                     | $65.8\pm 1.4$ | $83.9\pm 0.6$|
> > | **CIFAR (Animal vs. Not Animal)**  | $82 \pm 1.1$           | $87.7\pm 4.6$   | $88.8\pm 1.7$      | $91.1\pm 0.7$                     | $83.8\pm 1.3$ | $87\pm 1.1$  |
> > | **MNIST (Even vs. Odd)**           | $98.3 \pm 0.1$         | $97.7\pm 0.4$   | $97.8\pm 1.6$      | $98.1\pm 0.1$                     | $97.1\pm 0.4$ | $91.4\pm 0.5$|
> > | **Binarized Fashion MNIST**        | $89.6 \pm 0.2$         | $88.1\pm 1$     | $92\pm 1$          | $93.3\pm 0.75$                    | $90.7\pm 0.2$ | $86\pm 1.6$  |
> >
> > Appendix B contains additional notes for your reference.

---

> > > ### Comment · Reviewer_Wr1j · 2023-11-22
> > > **Thanks for the reply**
> > >
> > > Thanks for the additional experiments!
> > >
> > > For the first reply, I am still not convinced. What is the main difference between the proposed method and VQ-VAE? And why does the proposed method work well in the PU learning task, but VQ-VAE not?
> > >
> > > For the second, what are the experimental settings? Why are the experimental results of Robust-PU and Dist-PU different from ones in their original papers, such as ACC of Dist-PU on CIFAR-10 and Fashion-MNIST are 91.88, and 95.40, respectively?

---

> ### Author Response · Authors · 2023-11-22
> **Thanks for the response**
>
> > For the first reply, I am still not convinced. What is the main difference between the proposed method and VQ-VAE?
>
> Please note that the objective of a classical VQ-VAE is to learn a representation that can be used for image reconstruction which is a different goal that makes their loss function fundamentally different than the proposed loss function in this work.
>
> The loss function in the case of VQ-VAE [1] is $\mathcal{L} = \log p(x|z_q(x)) + \|\text{sg}[z_e(x)] - z_q(x)\|_2^2 + \beta \|z_e(x) - \text{sg}[z_q(x)]\|_2^2$
>
> where $\log p(x|z_q(x))$ represents the reconstruction loss, $\|\text{sg}[z_e(x)] - z_q(x)\|_2^2$ is the commitment loss, and $\|z_e(x) - \text{sg}[z_q(x)]\|_2^2$ is the vector quantization term.
>
> This loss function is directly designed to learn a representation that is suitable for image reconstruction, which is not necessarily suitable for sufficiently separating positive from negative samples to allow learning from PU data.
>
> The proposed method, on the other hand, is specifically tailored to learn from PU data; It is designed to project all positive samples to a vector while projecting the negative samples to a different vector, resulting in a separation between the two classes in the learned representation that facilitates learning a binary classifier. This is achieved by learning to minimize the loss function presented in Eq.(1) in the paper. The provided theorem in the paper in section 3.2 shed light on why the proposed idea is effective in achieving the separation betwen the positive and negative classes in the learned representation. This key separation between classes is not targeted when learning the representation of VQ-VAE.
>
> The proposed loss function in this work (Eq. (1) in the text) is:
> $\mathcal{L}(\mathbf{\\theta}) = \\sum_{i_{p}=1}^{n_P} \\sum_{j=1}^{K} \\|\\mathbf{v_{j}(x_{i_{p}};\\theta)} - \\text{sg}(\\mathbf{c_{m}})\\|^2_2 +  \\|\\text{sg}(\\mathbf{v_{j}(x_{i_{p}};\\theta)}) - Q(\\mathbf{v_{j}(x_{i_{p}};\\theta)})\\|^2_2 + \\sum_{i_{u}=1}^{n_u} \\sum_{j=1}^{K} \\|\\mathbf{v_{j}(x_{i_{u}};\\theta)} - \\text{sg}(\\mathbf{c_{1}})\\|^2_2 + \\|\\text{sg}(\\mathbf{v_{j}(x_{i_{u}};\\theta)}) - Q(\\mathbf{v_{j}(x_{i_{u}};\\theta)})\\|_2^2
> $
>
> The difference between the two loss functions is the main reason causing the difference in suitability for PU learning which is reflected in Figure 2 in the paper.
>
> In summary, the objectives of VQ-VAE and the proposed method are different and the loss functions are specifically designed to achieve their respective objectives.
>
> > For the second, what are the experimental settings? Why are the experimental results of Robust-PU and Dist-PU different from ones in their original papers, such as ACC of Dist-PU on CIFAR-10 and Fashion-MNIST are 91.88, and 95.40, respectively?
>
> Thanks for the question. The difference in accuracies is different as we are using different number of samples and different definition of the labeled classes: In CIFAR-10 dataset, we are learning from a dataset of size 38000 samples (compared to 51000 in Dist-PU paper and 60000 in Robust-PU) where the labeled set are 6 “animal” classes (Both of Dist-PU and Robust-PU use 4 vehicle classes). In Fashion MNIST dataset, we are learning from a dataset of size 38000 samples (compared to 60500 in Dist-PU paper and 70000 in Robust-PU) where the labeled set are the classes Sandal, shirt, sneaker, bag, and ankle boot (Each of Dist-PU paper and Robust-PU choose a different arbitrary split of classes). It is worth mentioning that the results shown in Dist-PU paper are for the case where $\alpha=0.4$ for both datasets, while our results table is for the case where $\alpha=0.5$. Our preliminary results on varying $\alpha$ (which can be found in appendix B.2) does not currently include the method Dist-PU or Robust-PU, but we are working on adding all of the remaining methods to the final version of the paper. We followed the same experimental settings used in the “Observer-GAN” paper to compare all the methods and mentioned that in the paper. We will add an appendix section for the final version detailing all the experimental details for clarity. Thanks for bringing that to our attention.
>
> [1] Van Den Oord, Aaron, and Oriol Vinyals. "Neural discrete representation learning." Advances in neural information processing systems 30 (2017).

---

### Official Review · Reviewer_NPjv · 2023-11-01

**Soundness:** 2 fair
**Presentation:** 2 fair
**Contribution:** 2 fair
**Rating:** 5
**Confidence:** 4

**Summary:**

This paper presents a simple method to solve the problem of learning a binary classifier with positive and unlabelled data. The proposed method is based on vector quantiza- tion technique to perform dimension reduction first, and then apply standard k-means algorithm to cluster the unlabelled data into positive and negative 2 clusters. In addition to the experimental evaluation, the paper provides some math intuition and ablation study to support and explain how the proposed method works. In the experiment section, the paper shows that the proposed method can produce comparable results w.r.t state-of-the-art GAN based methods.

**Strengths:**

The paper is well written and easy to understand. The results sound good and perhaps easy to re-produce if the authors can publish their code.

**Weaknesses:**

1. The idea is simple and the novelty may not be strong enough to publish in such a high standard conference.
2. The k-means algorithms need to keep running in each iteration. Although the idea is simple, it will be very slow if the data size is huge.
3. The proposed method is not convinced to handle the case when the labels are imbalanced.

**Questions:**

1. According to figure 4, the proposed method seems to fall into an interesting situation where the validation is good but the center of two clusters are closer after more epochs. Can author explain the reason?
2. Can the proposed method handle imbalanced labelled data? This happens in many real situations, such as CTR prediction. Typically clicks are much less than impressions. However, there will be lots of inventory that may not become impressions and therefore there is no label associate to it.
3. Would the proposed algorithm sensitive to the initialization of the cluster center?

---

> ### Author Response · Authors · 2023-11-21
>
> We appreciate the valuable feedback and suggestions. Here we try to address the concerns and suggestions:
>
> > The idea is simple and the novelty may not be strong enough to publish in such a high standard conference.
>
> Although the intuition behind the proposed method is simple, this simplicity contributes to its effectiveness and ease of implementation, which can both be seen as significant advantages. The main novelty in the proposed method lies in the idea of learning a vector-quantized representation space from PU data that separates positive and negative samples in the unlabeled set, which facilitates learning a binary classifier. To the best of our knowledge, this is the first work that introduces this idea. The introduction section further clarifies the main contributions presented in this work, with the results highlighting their significance and the theoretical justifications elucidating their soundness and validity.
>
> > The k-means algorithms need to keep running in each iteration. Although the idea is simple, it will be very slow if the data size is huge.
>
> We tested the speed of the K-means algorithm when training on different datasets, and added the results in section B.1 in the appendix. The results show that the K-means algorithm is relatively fast when running on the presented datasets. Although increasing data size will increase the time of running the algorithm, we expect the time to only increase linearly as the time complexity of K-means is linear in dimensions and number of samples. It should be pointed out that the Kmeans algorithm in the proposed method is run on the learned representation space, which is in lower dimensions than the input data.
>
> > The proposed method is not convinced to handle the case when the labels are imbalanced.
>
> Thank you for the valuable suggestion. Testing the efficacy of the proposed method compared to other methods in different data configurations is indeed a valuable study that we are now working to include in the final version. We ran an experiment to test the efficacy of the method when the positive samples are less or more likely than negative samples (different $\alpha$ values) and compared the results to the method $TED^{n}$ in Table 5 in section B.2 in the appendix. Given the limited time available for the review period and the multiple methods and data configurations to be tested, we were unable to include a comprehensive table covering all methods and configurations for now. We are currently compiling these results and plan to include them in the appendix of the paper in the final version.
>
> > According to figure 4, the proposed method seems to fall into an interesting situation where the validation is good but the center of two clusters are closer after more epochs. Can author explain the reason?
>
> Figure 4 in the paper shows that as the training proceeds, the distance between the two identified clusters decreases. It is noteworthy that in Figure 4, the distance between the clusters after epoch 40 is 2000. While this number is relatively smaller than the peak distance in the figure, there is no obvious reason to believe that this distance is not sufficient for clustering. It is obvious that if the distance keeps decreasing due to overfitting, it will reach a point where clustering will be meaningless; that's why we observe the distance between the clusters during training to pick the point at which the distance is at its peak.
>
> > Can the proposed method handle imbalanced labelled data? This happens in many real situations, such as CTR prediction. Typically clicks are much less than impressions. However, there will be lots of inventory that may not become impressions and therefore there is no label associate to it.
>
> Thank you for the suggestion. We provide a simple comparison between the proposed method and the method $TED^{n}$ in Table 5 in section B.2 in the appendix, and we are working on adding more extensive experiments to the appendix for the final version.
>
> > Would the proposed algorithm sensitive to the initialization of the cluster center?
>
> Although the initialization of the codebook vectors in the proposed method and the initialization of the cluster centers in the K-means algorithm can have an effect on the final performance of the method, the standard deviations of the proposed method in the results table show that it is robust to the random initialization of both the codebook vectors and the K-means cluster centers. The ablation studies section shows that learning the codebook vectors ensures the convergence to vectors that are suitable for the K-means algorithm. And observing the distance between the centers of the two identified clusters by the K-means algorithm (as shown in Figure 4 in the paper) ensures that the algorithm is identifying clusters that are consistently distanced from each other, which indicates that initialization of the cluster centers is proper.

---

> > ### Author Response · Authors · 2023-11-21
> > **Tables**
> >
> > Here are the thables we added to the appendix and referred to in the comment:
> >
> > The K-means algorithm speed per iteration (in seconds):
> >
> > | **Dataset**                          | **MNIST**            | **Fashion MNIST**    | **AFHQ**            |
> > |--------------------------------------|----------------------|----------------------|---------------------|
> > | **# of Samples, # of dimensions**    | $19000, 49$          | $19000, 49$          | $3300, 1024$        |
> > | **K-means Elapsed Time (s)**         | $0.0207 \pm 0.0038$  | $0.0292 \pm 0.0174$  | $0.3681 \pm 0.0071$ |
> >
> > Comparing the performance of the method to $TED^{n}$ for different $\alpha$ values:
> >
> > |                     | **α = 0.2**        | **α = 0.4**        | **α = 0.5**         | **α = 0.6**        | **α = 0.8**        |
> > |---------------------|--------------------|--------------------|---------------------|--------------------|--------------------|
> > | **TED^n**           | $86.9 \pm 1$       | $89.5 \pm 2.5$     | $89.9 \pm 14.9$     | $81.5 \pm 3.7$     | $71.1 \pm 2$       |
> > | **VQ K-means**      | $97.6 \pm 1.1$     | $96.5 \pm 0.7$     | $95.4 \pm 1.3$      | $84.3 \pm 1.1$     | $70.2 \pm 2.1$     |
> >
> > Appendix B contains additional notes for your reference.